



# QUANTIFYING SEASONAL CORNICE DYNAMICS USING A TERRESTRIAL LASER SCANNER IN SVALBARD, NORWAY

Holt Hancock[1,2], Markus Eckerstorfer[3], Alexander Prokop[4,5,1], Jordy Hendrikx[6]

[1]Department of Arctic Geology, University Centre in Svalbard, N-9171 Longyearbyen, Norway
[2]Department of Geosciences, University of Oslo, N-0371 Oslo, Norway
[3]Earth Observation Group, NORCE, N-9294 Tromsø, Norway
[4]Department of Geodynamics and Sedimentology, University of Vienna, 1090 Vienna, Austria
[5]Snow Scan GmbH, Research, Engineering, Education, Stadlauerstrasse 31, 1220 Vienna, Austria
[6]Snow and Avalanche Lab, Department of Earth Sciences, Montana State University, 59717 Bozeman, Montana, USA

*Correspondence to*: Holt Hancock (holt.hancock@unis.no)

**Abstract.** Snow cornices develop along mountain ridges, edges of plateaus, and marked inflections in topography throughout regions with seasonal and permanent snow cover. Despite the recognized hazard posed by cornices in mountainous locations, limited modern research on cornice dynamics exists and accurately forecasting cornice failure continues to be problematic. Cornice failures and associated cornice fall avalanches comprise a majority of observed avalanche activity and endanger human life and infrastructure annually near Longyearbyen in central Svalbard, Norway. In this work, we monitored the seasonal development of the cornices along the plateaus near Longyearbyen with a terrestrial laser scanner (TLS) during the 2016/2017 and 2017/2018 winter seasons. The spatial resolution at which we acquired snow surface data with TLS enabled us to observe and quantify changes to the cornice systems in detail not previously achieved. We focused primarily on the evolution and failure of the lower cornice surfaces where accessibility has precluded previous research. We measured cornice accretion rates in excess of 10 mm hr$^{-1}$ during several accretion events coinciding with winter storms. We observed five cornice fall avalanche events following periods of cornice accretion and one event following a warm period with mid-winter rain. The results of our investigation provide quantitative reinforcement to existing conceptual models of cornice dynamics and illustrate cornice response to specific meteorological events. Our results demonstrate the utility of TLS for monitoring cornice processes and as a viable method for quantitative cornice studies in this and other locations where cornices are of scientific or operational interest.



## 1 Introduction

Snow cornices are overhanging projections of snow forming due to the deposition of wind transported snow in the lee of ridgelines or sharp slope inflections (Montagne et al., 1968;Seligman, 1936). Cornices have attracted interest for their hydrologic implications (e.g. Anderton et al., 2004) and as agents of geomorphic change in periglacial environments (Eckerstorfer et al., 2013;Humlum et al., 2007), but are perhaps best recognized as a snow and avalanche hazard in mountainous terrain (Montagne et al., 1968;Vogel et al., 2012). Cornices pose an avalanche hazard when they fail either as a

full cornice failure with the entire cornice detaching from the ground or as a partial failure with a smaller cornice mass separating from the rest of the cornice. The detached cornice blocks travel downslope under the influence of gravity and become a cornice fall avalanche by entraining loose surface snow or triggering a snow slab on the slope below (e.g. Vogel et al., 2012). In ski areas or where cornices and cornice fall avalanches endanger infrastructure, both explosives (Farizy, 2013;McCarty et al., 1986) and structural defenses (e.g. Montagne et al., 1968) are employed operationally to mitigate

cornice hazard. Most cornice-related fatalities, however, occur in recreational backcountry settings and result from the victim's weight triggering cornice failure.

Despite the well-recognized hazards and operational challenges associated with cornices and cornice fall avalanches, specific cornice research is relatively scarce. Early cornice studies summarized by Vogel et al. (2012) focused on qualitative descriptions of cornice formation processes and resulting cornice structures (e.g. Montagne et al., 1968;Seligman, 1936).

Later studies investigated mechanisms by which individual snow crystals adhere during cornice accretion (Latham and Montagne, 1970), the physical snow characteristics at various structural locations on individual cornices (Naruse et al., 1985), and the specific interactions between wind-drifted snow and cornice morphology during cornice formation (Kobayashi et al., 1988).

Recent work has refined the conceptual model of seasonal cornice dynamics established by these earlier studies primarily by

employing time-lapse photography to examine cornice responses to the meteorological factors controlling the development and failure of cornices (Munroe, 2018;van Herwijnen and Fierz, 2014;Vogel et al., 2012). Vogel et al. (2012) observed cornice processes over two winter seasons on a single mountain slope in central Svalbard and proposed a conceptual model of seasonal cornice dynamics including cornice accretion, deformation, and failure. Their results indicated cornice accretion occurs during or immediately following winter storms with wind speeds in excess of 10 m s$^{-1}$ from a direction perpendicular

to the ridgeline, while cornice scouring resulted from strong winds oriented towards the cornice's leading edge. Smaller cornice failures observed in this study were clustered in June near the end of the snow season and coincided with increasing air temperatures and decreased snow strength. Less frequent failures in the earlier part of the winter often involved the entire cornice mass and resulted in the some of the largest cornice fall avalanches observed in the study.

Later work in an alpine setting also linked cornice accretion to strong winds during or soon after a snowfall and found the

SNOWPACK wind drift index correlated well with cornice width estimates (van Herwijnen and Fierz, 2014). No cornice

failures or cornice fall avalanches were observed in this study, however. Munroe (2018) used time-lapse photography to observe the growth and repeated failure of a cornice in Utah, USA. He also found cornice accretion to primarily coincide with periods of snowdrift. He divided the nineteen cornice fall avalanches observed in his study into two distinct groups: snow-caused cornice fall avalanches where failure primarily resulted from snow loading on the cornice and temperature-caused avalanches where failure is related to rapid temperature increases presumably leading to destabilization of the cornice through the loss of snow strength.

We build upon the observational understanding and conceptual model of seasonal cornice dynamics established in these previous works by monitoring cornice systems in Longyeardalen – including one site previously examined by Vogel et al. (2012) – with a terrestrial laser scanner (TLS). TLS – or ground-based LiDAR (Light detection and ranging) – is an active remote sensing technology with documented applications for observing and monitoring various slope processes and hazards including landslides (Jaboyedoff et al., 2012;Prokop and Panholzer, 2009), coastal cliff erosion (e.g. Caputo et al., 2018), and rock slope instability (Abellán et al., 2014). TLS is being increasingly employed in snow and avalanche research to map snow depth and snow depth change (e.g. Deems et al., 2013;Fey et al., 2019;Prokop, 2008;Schirmer et al., 2011). Other specific snow-related applications include quantifying snow drift processes to verify physical models (Mott et al., 2011;Schön et al., 2015;Vionnet et al., 2014), observing avalanche activity to calibrate dynamic avalanche models (Prokop et al., 2015), assisting avalanche control operations (Deems et al., 2013), and as a tool in planning snow drift control fences (Prokop and Procter, 2016).

We monitored cornice accretion, deformation, failure, and associated cornice fall avalanche activity near Longyearbyen, Svalbard with TLS technology over two winter seasons (2016/2017 and 2017/2018). To our knowledge TLS has not been employed to specifically monitor cornice dynamics, so our primary objectives are to use the high spatial resolution snow surface data acquired via TLS to:

1. Demonstrate the utility of TLS to observe cornice processes
2. Observe and quantify cornice accretion, deformation, failure, and associated cornice fall avalanches and link these processes to their controlling meteorological factors.
3. Use our findings to provide suggestions for forecasting cornice fall avalanches in this and other locations threatened by cornices.

## 2 Study Area

*Figure 1 here*

The present study focuses on the cornices forming above the Longyear valley (hereafter: Longyeardalen) in central Svalbard (Figure 1). Longyeardalen is a glacially sculpted, u-shaped valley with a northeast/southwest oriented valley axis running



approximately 3 km from the termini of two small mountain glaciers to a fjord. The Gruvefjellet and Platåberget plateaus border Longyeardalen to the west and east, respectively, with Svalbard's administrative center, Longyearbyen, situated in the valley bottom. The Gruvefjellet and Platåberget slopes lie within the horizontally-bedded, lower-Tertiary aged Van Mijenfjord Group of sandstones and shales (Major, 2001). Resistant strata within this group form the area's extensive

plateau topography. The entire region is underlain by continuous permafrost (Humlum et al., 2003).

*Figure 2 here*

We investigated seasonal cornice dynamics and cornice fall avalanches along and under the Gruvefjellet and Platåberget plateau margins, respectively (Figure 2). The steep valley walls descending from the broad plateau summits (approximately 450 m elevation) are characterized in their upper portions by protruding resistant bedrock buttresses and transport couloirs

incised by fluvial and gravitational slope processes. The Gruvefjellet slope described in detail by Eckerstorfer et al. (2013) consists of a 50-70 m near-vertical bedrock cliff situated between under the plateau margin and over a 40-50° slope that serves as a slab avalanche release area. This broad slope transitions into the transport couloirs which in turn feed extensive avalanche fan deposits downslope. Similar morphology exists on the Platåberget slope, but the plateau margin transitions directly into discrete 45-55° release areas leading into the couloirs and lacks the near-vertical bedrock face present on

Gruvefjellet.

Central Svalbard's climate is cold and arid, with a mean annual air temperature of -4.6°C and mean annual precipitation of 191 mm at the Svalbard Airport automated weather station (AWS) for the 1981-2010 period of record (Førland et al., 2011). Combined mean winter (DJF) and mean spring (MAM) precipitation for 1981-2010 is 86 mm (Førland et al., 2011). Mean winter air temperature for the same period is -11.7°C and mean spring air temperature is -8.3°C (Førland et al., 2011).

Rapidly increasing air temperatures in the winter and spring in response to decreased sea ice extent (Isaksen et al., 2016) create difficulties establishing representative baseline temperature conditions, with recent reports indicating warming on the order of 3-5°C for Svalbard as a whole from 1971 to 2017 (Hanssen-Bauer et al., 2019). Less clear changes are apparent in the precipitation trends (e.g. Hanssen-Bauer et al., 2019), but mid-winter rain-on-snow events are dramatically increasing in frequency (e.g. Vikhamar-Schuler et al., 2016).

Svalbard's climatic situation prohibits the growth of woody vegetation, and snow distribution across the landscape is strongly controlled by the wind (e.g. Jaedicke and Sandvik, 2002). Southeasterly winds generally prevail across the region's plateau mountains, but often switch to westerly or southwesterly during winter storms and are frequently redirected along the major valley axes at lower elevations (Christiansen et al., 2013). Winter weather in central Svalbard fluctuates between extended periods of cold, stable high pressure punctuated by warm, wet low pressure systems conveyed northwards along

the North Atlantic cyclone track (Hanssen-Bauer et al., 1990;Rogers et al., 2005). This is reflected in the region's snow and avalanche climate, where the snowpack typically consists of persistent weak layers formed during high pressure interspersed with wind slabs or ice layers formed during snow storms or rain-on-snow events (Eckerstorfer and Christiansen, 2011a).

Avalanche activity here displays a strong topographical and meteorological control, with direct action slab avalanches clustered around winter storms and the region's plateaus serving as source areas for the extensive cornice systems that
contribute to frequent cornice fall avalanches (Eckerstorfer and Christiansen, 2011c).

## 3 Methods

### 3.1 Automated snow and weather data

We obtained wind and air temperature data from the Gruvefjellet automated weather station (AWS), precipitation data from the Svalbard Airport AWS, and a limited time series of snow depth data from a pair of ultrasonic snow depth sensors placed
in avalanche release areas on Gruvefjellet and Platåberget during the 2017/2018 winter season (Figures 1 and 2). We defined the winter season as 1 December to 30 June for the purposes of this study. The Gruvefjellet AWS is located less than 500 m east of the Gruvefjellet cornice system at an elevation 464 m and records hourly meteorological data. The Svalbard Airport AWS is situated approximately 5 km northwest of the study area at 28 m and is the only weather station in the region with long-term precipitation measurements.

As part of the installation of a network of automated snow monitoring stations in Longyeardalen (Prokop et al., 2018) we mounted two ultrasonic snow depth sensors in avalanche release areas under the cornice systems in autumn 2017. These sensors were located at 350 m and 450 m elevation on Gruvefjellet and Platåberget, respectively (Figure 2). We employed the Campell Scientific SR50A ultrasonic distance sensor to measure snow depth at each location. The snow sensors began recording reliable snow depth data on 15 November 2017 and continued until the end of the 2017/2018 season at ten-minute
intervals.

### 3.2 Terrestrial laser scanning (TLS) and post-processing

We used a Riegl® Laser Measurement Systems VZ-6000 ultra-long range terrestrial laser scanner to repeatedly scan the Gruvefjellet and Platåberget cornice systems throughout the 2016/2017 and 2017/2018 winter seasons. The VZ-6000's 1064 nm operating wavelength is particularly well-suited for measuring snow surfaces, while the high scanning speed and
measurement range up to 6 km with a 30 kHz pulse repetition rate ensured adequate data acquisition capabilities across the study area in a variety of atmospheric conditions (Riegl, 2019;Prokop, 2008).

We use data from 25 scans of Gruvefjellet and 22 scans of Platåberget during the duration of the study (Appendix I). Of these, one scan from each Gruvefjellet and Platåberget is a snow-free surface taken 16 September 2016. For Gruvefjellet, we acquired usable snow surface data from 18 scans during the 2016/2017 season and 7 scans during 2017/2018. We acquired
14 snow surface scans of Platåberget during 2016/2017 and seven scans during 2017/2018. The TLS was unfortunately damaged in late April 2018 and we were unable to acquire any scans after our final scan on 13 April 2018.



We pre-processed the raw point clouds in RiSCAN Pro, Riegl's proprietary data processing software. We established a suite of ground control points on both Platåberget and Gruvefjellet using a differential global positioning system (DGPS) which we used to georeferenced individual point point clouds. We then aligned repeated snow-covered scans to the snow-free scans
established in September 2016 using these ground control points and the "Multi Station Adjustment" plugin in RiSCAN Pro following the approach outlined by Prokop and Panholzer (2009). The 3D accuracy of this registration process ranged from <1 mm to 70 mm (Appendix 1). We then manually filtered non-ground points or points above the snow surface. Finally, we resampled the resulting point clouds to a 0.10 m grid and exported to an XYZ text file.

We imported individual point clouds into CloudCompare (CloudCompare, 2019) , for further analyses (Figure 3). To create
2D cornice profile cross-sections, we extracted point cloud profile sections along manually defined axes using the polyline extraction tool native to CloudCompare (Figure 3c,d). This tool requires user-defined inputs for profile type, section thickness, and maximum edge length which we set to "Both", 0.6 m, and 0.2 m, respectively. We then manually edited and digitized the resulting shapefiles in the ArcScene 3D Editing environment (ArcGIS 10.4.1) to create the vertical cornice profile schematics as 3D shapefiles.

We calculated representative volumes for selected areas from both the Platåberget and Gruvefjellet cornice systems using the Compute 2.5D Volume tool in CloudCompare. This tool computes the volume between two 2.5D point clouds by rasterizing the point clouds to a specified grid size and then computing volumes based on the differences in a specified projection direction between the rasterized values (Figure 3 e,f,g,h). In our case, we rasterized our point clouds to a 1 m grid and calculated horizontal distance differences along the "X" projection direction, which in our georeferenced point clouds
corresponds to east-west (i.e. the slope fall-lines). For each cornice system, we computed the volume of snow in 40 m x 8 m area of the plateau margin for each usable snow surface scan by subtracting the bare-earth surface from the scanned snow surface (Figure 3 e,f,g). We chose this areal extent to maximize coverage of an individual cornice throughout its development during the season (i.e. to completely capture the vertical extension of the leading edge) while minimizing volume changes related to other snow on the slope.

*Figure 3 here*

We used the Multiscale Model to Model Cloud Comparison (M3C2) algorithm developed by Lague et al. (2013) and implemented as a plugin in CloudCompare to quantify changes to the cornices and snow surfaces on the slopes below in 3D. The M3C2 algorithm allows for direct comparison of point clouds in 3D and is specifically developed to handle 3D differences and detect changes to complex surfaces where both vertical and horizontal changes exist (Lague et al., 2013).
This functionality requires the user to input the following parameters: the normal scale, the projection scale, and the maximum depth (e.g. Lague et al., 2013;Watson et al., 2017). We selected a normal scale of 2 m oriented positively to the scan position (i.e. the normals "face" the scan position), a projection scale of 1 m, and a maximum depth of 10 m for all M3C2 calculations.



TLS-based snow surface measurement accuracy generally decreases with increasing distance from the scanner to the
measured snow surface and is affected by the manner in which the laser beam interacts with the snow surface, the local
terrain characteristics, the stability of the scanner while scanning, and the quality of the scan data registration process (Fey et
al., 2019;Hartzell et al., 2017;Prokop et al., 2008). The relative accuracy – the deviation between measurements of an
unchanged surface taken under different measurement conditions – can be assessed to quantify uncertainties related to both
registration errors and positional errors from the interaction of the laser beam with the surface (Fey et al., 2019;Prokop and
Panholzer, 2009). We assessed relative accuracy for our data by measuring M3C2 distances between each snow covered
scan and the snow-free scan on a 10x10 m area of stable, snow-free rock faces near the cornices on both Gruvefjellet and
Platåberget (Figure 2). We report relative accuracy for each snow-covered scan as the mean of all M3C2 distances on the
10x10 m area (Appendix 1) Locations for relative accuracy assessment were selected based on their ability to remain
functionally snow-free throughout the study and because they were not otherwise used in the registration process. As both
registration and positional errors can be spatially variable across the scanned area (Fey et al., 2019;Hartzell et al.,
2017;Prokop, 2008), we used these locations in close proximity to the cornices of interest to best represent the relative
accuracy near the cornices. Mean M3C2 distance values

are smaller than 80 mm for all scans, with standard deviations ranging from <10 mm to 72 mm (Appendix 1). Uncertainty
associated with the relative volume metric, calculated by multiplying the relative accuracy of each scan by the surface area
considered in the volume calculations (369 $m^2$) thus ranged from less than 1 $m^3$ to 28 $m^3$ (Appendix 1).

### 3.3 Supplemental Observational Data

We relied on snow and avalanche observations from Platåberget and Gruvefjellet from the Norwegian Water Resources and
Energy Directorate's (NVE) online observation platform regObs ([www.regobs.no](www.regobs.no)) to supplement our TLS observations.
Local observers conduct snow and avalanche assessments on the Gruvefjellet and Platåberget slopes on a sub-weekly basis,
so we were able to much better constrain avalanche cycle timing than with the temporal resolution available from the TLS
data.

## 4 Results

### 4.1 Seasonal summaries

### 4.1.1 2016/2017

We compare seasonal meteorological conditions (Figure 4) with cross-sectional cornice profiles derived from eight scanned
snow surfaces on Gruvefjellet and seven surfaces on Platåberget. We selected these profiles from a pool of 18 usable scans
from Gruvefjellet and 14 from Platåberget (Appendx 1) to represent key points in the development of the cornice systems.



Small cornices had accumulated on Gruvefjellet by 2 December 2016. Maximum horizontal cornice growth prior to this scan occurred in the vicinity of profile GF2, where both vertical and horizontal cornice growth exceeded 1 m from the edge of the

plateau (Figure 5). The representative cornice volume in the vicinity of profile GF1 already approached 200 m$^3$. Temperatures remained below freezing over the next month, and daily averaged wind speeds exceeded 10 m s$^{-1}$ only on 29 December 2016. By 12 January 2017, the representative cornice volume on Gruvefjellet had more than tripled relative to early December to over 600 m$^3$. Horizontal cornice extension along the Gruvefjellet cornice system exceeded 4 m in most locations, with maximum horizontal extension near profile GF1 exceeding 5 m (Figure 5). The representative cornice

volume of just over 300 m$^3$ from the Platåberget cornices on the same date show considerably less cornice growth (Figure 4).

*Figure 4 here*

Heavy snowfall followed by strong westerly winds preceded several cornice fall avalanches on 21 January on Platåberget (Figure 4, Table 1). Representative cornice volume on Platåberget nearly doubled from roughly 300 m$^3$ to over 600 m$^3$ between the 12 January and 21 January scans. Horizontal accretion on profile PB2 exceeded 3.5 m, resulting in an accretion

rate of 17 mm hr$^{-1}$ (Table 2). The cornice represented by profile PB1 failed, triggering a cornice fall avalanche (Size D2, R3) which reached the road at the foot of the slope. The failure plane represented by the 21 January profile on PB1 does not extend back to the 12 January surface, suggesting newly accreted snow comprised the bulk of the failure (Figure 5). Cornices on Gruvefjellet experienced comparably minor changes during this event, with the representative volumes decreasing by just 30 m$^3$ and minimal changes evident in the profiles (Figure 5).

*Table 1 here*

A major accretion event in mid-February 2017 followed several weeks of unseasonably high temperatures at cornice elevation during early February (Figure 4). Locally heavy snowfall and strong easterly winds accompanying a vigorous winter storm impacted the region between 19 February and 21 February. Profile GF1's horizontal extension increased by nearly 3 m between the 17 February and 24 February scans resulting in horizontal accretion rates exceeding 15 mm hr$^{-1}$

(Table 2). The representative volume increased by approximately 100 m$^3$ during the same timeframe. The strong easterly winds stripped the Platåberget cornice system on the windward side of the valley reflected by the abrupt decrease of over 100 m$^3$ to the representative volume there.

*Figure 5 here*

Representative volumes for both cornice systems gradually increased in the following month, and profiles from 21 March

2017 show considerable rounding and downslope creep of the cornices' leading edges in profiles GF1 and GF3 (Figure 5). Cornices continued to grow on Platåberget, with horizontal growth exceeding 2 m on portions of the PB1 and PB2 profiles and PB3's vertical extent increasing by over 2 m. The Platåberget cornices did not deform downslope to the same degree as the Gruvefjellet cornices during this time period. A representative volume decrease of over 500 m$^3$ (roughly 50% of the





volume) on Gruvefjellet in April is related to a major cornice failure near profile GF1, while Platåberget's representative
volume increased by 150 m³ in an accretion event near the end of the month (see Section 4.2.1 ). Considerable cornice
accretion is evident in all cornice profiles between 21 March and 1 May except for profiles GF1 and PB2 where we
documented cornice failures. Representative volumes continued to increase in early May as light precipitation coincided with
continued subzero temperatures. Representative cornice volume on Gruvefjellet gradually increased through 31 May and
then dramatically decreased with the onset of sustained positive temperatures at the Gruvefjellet AWS. Cornices on
Platåberget continued to accrete through the 18 May scan before beginning to melt away between 18 May and 9 June.

### 4.1.2 2017/2018

*Figure 6 here*

We gathered seven scanned snow surfaces from both Gruvefjellet and Platåberget for the 2017/2018 season with which to
compare to meteorological conditions. Cornice development during the 2017/2018 winter season differed considerably from
2016/2017 despite relatively similar seasonal meteorological conditions (Table 3). Gruvefjellet profiles from 15 December
2017 show over of 5 m horizontal cornice growth in all profiles, and representative volume approached 1000 m³
(Figures 6 and 7). Contrastingly, the Platåberget plateau margin remained functionally free of snow. Cornices continued to
grow over the following five weeks on Gruvefjellet up to the 24 January scan, with profiles GF1 and GF3 reaching their
maximum horizontal extensions for the season of nearly 7 m and over 8 m, respectively (Figure 7). Cornice fall avalanches
260    observed on 13 January 2018 are evident in the decreased cornice extension in GF2 in the 24 January scan relative to the 15
December 2017 surface and were associated with positive air temperatures and rain at cornice elevation (Table 1). Profiles
on Platåberget on 24 January 2018 do not show cornice development, with snow accumulating relatively parallel to the
underlying topography.

*Table 3 here*

265    Representative volume doubled on Platåberget between the 31 January and 22 February scans from 400 m³ to 900 m³. This
coincided with a 0.34 m increase in snow depth at the snow sensor during a snowstorm on 5 February and 6 February where
14 mm of precipitation was measured at the Svalbard Airport AWS (Figure 6). Cornice system changes were more minimal
on Gruvefjellet, with a subtle increase of 100 m³ in representative volume. Measured snow depth on Gruvefjellet increased
from 1.45 m on 31 January to a maximum of 1.77 m on 13 February, before slowly decreasing back to 1.48 m by 22
February (Figure 6). A minor decrease in horizontal extension (<1 m) and slight downslope deformation exhibited in profile
GF1 are the main observed changes to the cornice profiles between 31 January and 22 February (Figure 7).

Snow depths increased by 0.20 m and 0.28 m on Gruvefjellet and Platåfjellet, respectively, on 26 and 27 February 2018 as
over 7 mm precipitation was recorded at the airport (Figure 7). A marked increase in representative volume of 230 m³ on
Platåberget between the 22 February and 2 March scans coincides with an increase in snow depth of 0.28 cm over 26 and 27



February. Although a small cornice is evident in profile PB3 on 2 March, increased volume during this time illustrates slope-normal snow depth increase rather than cornice accretion in the representative volume area in the vicinity of PB2 (Figure 7). On Gruvefjellet, downslope creep of the cornice masses continued, with maximum vertical deformation close to 0.80 m for the leading edge of profile GF1. A winter storm on 18 March 2018 resulted in cornice failures seen in both GF1 and GF2 and decreased representative volume on Gruvefjellet, while scouring reduced volume during this time on Platåberget (see

Section 4.2.2). Minimal further changes are evident in season's final scan (due to scanner failure) taken 13 April 2018.

*Figure 7 here*



## 4.2 Case Studies

### 4.2.1 April 2017

*Figure 8 here*

We documented three periods of cornice fall avalanche activity with TLS data in April 2017. In the first, a small portion of the cornice between profiles GF1 and GF2 failed on 9 April following a period of precipitation falling as snow and easterly winds in excess of 10 m s$^{-1}$ (Figure 8a, Figure 9a annotation 1). The cornice represented by profile GF1 then failed completely on 21 April 2017 coincident with trace precipitation falling as snow and two days of moderate to strong easterly

winds (Figure 8b, Figure 10). Negative M3C2 distances displaying changes to the Gruvefjellet cornice system between the 21 March 2017 and 25 April 2017 scans show the largest portion of the failed cornice along the axis of profile GF1 (Figure 9a, annotation 2). This failure extended northwards almost 40 m along the cornice. Negative M3C2 distances on the vertical rock face immediately downslope of both the 9 April and 21 April cornice failures show how the falling cornice blocks remove snow from the rock face before impacting avalanche release areas below (Figure 9a, annotation 3). Here,

cornice impact craters and small slab avalanche releases are apparent in negative M3C2 distances (Figure 9a, annotation 4). Lower on the slope, the cornice fall avalanche deposition – complete with intact cornice chunks in the avalanche debris – is apparent in strongly positive M3C2 distances (Figure 9a, annotation 5). Other positive M3C2 distances along the cornices (Figure 9a, annotation 6) and horizontal and vertical extent increases on profiles GF2 and GF3 (Figure 10) show cornice accretion occurred elsewhere along Gruvefjellet during this time span. The easterly winds stripped the cornices on

Platåberget, evidenced by representative volume decreases of nearly 200 m$^3$ and vertical extension decreases of up to 1.5 m at profile PB3 (Figure 10).

*Figure 9 here*

A warm winter storm accompanied by 4.5 mm of precipitation, southwesterly winds, and air temperatures approaching 0°C at cornice level resulted in a period of major cornice accretion and associated cornice fall avalanche activity on the

Platåberget cornice system between the 25 April 2017 and 1 May 2017 scans (Figure 8c). Widespread cornice failures are shown by negative M3C2 distances along the Platåberget plateau margin (Figure 9b, annotation 1). These failures coincide with positive M3C2 distances in excess of 1.5 m indicative of cornice accretion elsewhere along the plateau margin (Figure 9b, annotation 2). Profile PB3, for example, experienced over a meter of increased vertical cornice extension (Figure 10) just south of a cornice failure shown in the M3C2 distances (Figure 9b, annotation 3). In profile PB1, 2 m

maximum increases in horizontal extension resulted in accretion rates of 17 mm hr$^{-1}$ (Table 2). The semi-vertical profile surface shown in profile PB2 (Figure 10) combined with the M3C2 distance decreases in the profile's immediate surroundings (Figure 9b, annotation 4). indicate cornice failure here occurred after some vertical cornice accretion, as the failure plane extends above the cornice roof from the 25 April snow surface. Cornice blocks released from this cornice and the one immediately to the north poured over cliffs further downslope and gouged impact craters (Figure 9b, annotation 5)



before releasing slab avalanches lower on the slope (Figure 9b, annotation 6). Minimal changes to the Gruvefjellet cornices occurred during this event.

*Figure 10 here*

### 4.2.2 March 2018

*Figure 11 here*

A storm in mid-March 2018 punctuated a month of otherwise stable weather and resulted in cornice fall avalanches on Gruvefjellet (Figure 11a). From 15 March to 19 March, 5.6 mm of precipitation accumulated at the Airport AWS, snow depths at the Gruvefjellet sensor increased by a maximum of 18 cm while those at the Platåberget sensor decreased by approximately 0.25 m, and strong winds blew from the ENE for 24 hours on 17 March – 18 March. Two large cornice failures on Gruvefjellet visible as strongly negative M3C2 distances near profile GF1 and slightly to the north (Figure 12a,

annotation 1) triggered avalanches on the slope below (Figure 12a, annotation 2). Similar to the morphology observed in the April 2017 cornice fall avalanches, the failed cornice blocks stripped snow off the vertical rock face and created impact craters while entraining snow as they moved downslope. The cornice chunks from these cornice failures also remained intact throughout the event and ran further than the rest of the avalanche debris (Figure 12a, annotation 3). A cornice block approximately 5 m in horizontal extension detached from the cornice represented by profile GF1, while a smaller (<1 m

horizontal extension) piece detached near GF2 (Figure 13). The GF3 profile did not fail, but over 1 m of snow accreted vertically on the cornice's leading edge. By contrast, Platåberget's plateau margin lost snow, with over snow depth decreases in excess of 0.20 m measured at the snow station and strongly negative M3C2 distances across the upper portion of the Platåberget release areas (Figure 12b, annotation 1).

*Figure 12 here*

*Figure 13 here*

## 5 Discussion

### 5.1 Seasonal cornice dynamics

TLS-derived cornice data from the 2016/2017 and 2017/2018 winter seasons provide quantitative reinforcement to the conceptual models of cornice dynamics developed in previous studies (e.g. Montagne et al., 1968;Vogel et al., 2012). In

these models, cornices accrete through relatively discrete events and begin to deform under their own weight before either failing or melting away towards the end of the snow season.

Our data show cornices can rapidly accrete at any point in the snow season given abundant snow available for wind transport, wind speeds sufficient to mobilize surface snow, and wind directions oriented relatively perpendicular to the



ridgeline. We documented accretion rates in excess of 15 mm hr$^{-1}$ at various times throughout the 2016/2017 season on each

side of the valley, which can be considered a minimum given the relatively poor temporal constraints on the accretion events

provided by the TLS data. Each of these periods of accretion coincided with measured precipitation at the airport, wind

speeds in excess of 5 m s$^{-1}$, and wind directions roughly placing the plateau margin in the lee. The relatively small proportion

of the winter seasons characterized by meteorological conditions conducive for accretion suggests just a few accretion events

play a key role in cornice development each season (Table 3). Asynchronous cornice responses on Gruvefjellet and

Platåberget to specific weather events further illustrate the importance of wind direction in controlling cornice dynamics.

During the February 2017 event, for example, cornices on Gruvefjellet rapidly accreted and gained volume with plentiful

snow available for transport and strong easterly winds. Cornices on Platåberget lost volume, however, as they were eroded

by the same easterly winds. Similar out of phase behavior was exhibited in late April 2017, when precipitation and westerly

winds resulted in considerable cornice growth on Platåberget accompanied by slight decreases to horizontal and vertical

extension in profiles GF2 and GF3 and minimal representative volume changes near profile GF1.

Following initial accretion, the cornices' leading edges begin to deform downslope. Deformation becomes more pronounced

later in the season, presumably as increased air temperatures and solar radiation begin to warm the snow, decreasing the

stiffness of the cornices and increasing creep (e.g. Schweizer et al., 2003). Further accretion events can then be

superimposed on this deformation as the season progresses, with short accretion events interspersed by longer periods of

downslope creep. This can be seen in the minor increases in horizontal extension and continued downslope deformation in

GF1 and GF3 through the latter portion of the 2017/2018 season (Figure 7). Cornice accretion and downslope deformation

can also occur almost simultaneously with air temperatures approaching or even exceeding freezing at cornice level, as

evidenced by the rapid accretion and downslope creep shown in profile PB1 for the 25 April – 1 May scan interval

(Figure 10).

While meteorological conditions control the specific timing of cornice accretion and downslope deformation, the underlying

topography appears to act as a fundamental control on cornice structure and seasonal cornice dynamics. The presence of the

steep bedrock face directly beneath the Gruvefjellet plateau margin limits the support provided by the underlying topography

compared to the gentler sloping Platåberget margin. The result is a more overhung cornice structure on Gruvefjellet, while

Platåberget's topography allows for more slope-normal snow accumulation. Profiles PB1 and PB2 failed to develop cornices

at all during the 2017/2018 winter season. The presence of cornices with horizontal extension approaching 5 m in these

locations during the 2016/2017 winter season, however, shows the topography can support cornice development given the

right meteorological conditions. Differences in meteorological conditions between the 2016/2017 and 2017/2018 winter

seasons may provide a partial explanation for differing seasonal snow cover responses on Platåberget (Table 3). Winds in

excess of 5 m s$^{-1}$ – a conservative estimate for threshold wind speeds required to mobilize loose snow (Li and Pomeroy,

1997) – from the western quadrant conducive to cornice accretion on Platåberget were slightly less prevalent during the

2017/2018 season, and these winds also coincided with precipitation roughly half as frequently as during the 2016/2017





season. Easterly winds exceeding 5 m s$^{-1}$ were considerably more prevalent during the 2017/2018 season which may have increased cornice scouring or limited snow available for transport –and thus accretion – on Platåberget. Nevertheless, the meteorological differences between the two winter seasons are subtle enough when compared to the noteworthy differences

in cornice dynamics to suggest specific interactions between meteorology and topography not necessarily captured by our analyses meaningfully impact cornice development.

Topography also seems to control the relative size of cornice failures. Vogel et al. (2012) describe a "geomorphologically determined sedimentary step approximately 3 m below the plateau that most likely acts as the cornice pivot point" on Gruvefjellet. This pivot point is most evident in profile GF1, where in both winter seasons the downslope creep of the

overhung cornice beyond this pivot point ultimately became overburdened during an accretion event and caused the cornice to fail completely. The cornice represented by GF1 has the least topographic support and developed the most overhanging cornice structure of the specific cornices we investigated, while also failing completely both seasons. By contrast, the topographic support provided by Platåberget does not promote overhanging cornices to the same degree, instead promoting a thicker slope-normal snowpack which in itself supports the cornice structure. Here, observed cornice failures such as that

shown in PB2 during the 25 April - 1 May 2017 scan interval (Figure 10) are limited to the recently accreted snow and did not involve the entire cornice mass. Similarly, profile GF2 failed in March 2018 within hours of profile GF1's full failure, but involved a much smaller portion of the cornice predating the 2 March scan – potentially related to increased topographic support to this cornice relative to GF1 (Figure 13).

**5.2 Cornice fall avalanches**

Previous work has differentiated cornice fall avalanche types by the inferred mechanism of cornice failure – either via increased snow load from accretion or decreased snow strength in the cornice related to increased snow and air temperatures. Five of the six cornice fall avalanche events observed in this study coincided with winter storms leading to accretion just prior to cornice failure (Table 1). This is in contrast to previous findings from this location, where no cornice failures were observed in direct response to snow loading caused by a snow storm (Vogel et al., 2012). The lone cornice fall avalanche

event we cannot link to cornice accretion occurred in January 2018. This event coincided with heavy precipitation, but positive temperatures at the Gruvefjellet AWS and decreasing snow depths at the Gruvefjellet snow sensor indicate this precipitation fell as rain (Figure 6). Our truncated TLS observation record in late-spring 2018 unfortunately omits the May-June period found by Vogel et al. (2012) to be critical for air temperature induced cornice failures in this location, but observational records throughout this time do not indicate further cornice fall avalanches. Accretion's role in determining

cornice failure is also reflected in the asynchronous timing of cornice failures on Gruvefjellet and Platåberget during our study. None of the observed avalanche events included activity on both Gruvefjellet and Platåberget simultaneously as would be expected with air temperature induced failures, with avalanches instead occurring only on the leeward aspect.



Observed cornice fall avalanche size appears to be controlled largely by the snow conditions in the underlying release area.
Cornice fall avalanches on Gruvefjellet follow a pattern exemplified by the April 2017 case study in which the cornice fails
and removes snow from the steep bedrock face below as it descends before impacting the release areas at the base of the cliff
(e.g. Figure 9a). The cornice block can then, depending on the snow conditions in the release area, entrain snow from its
impact crater and the avalanche path below or trigger a larger slab avalanche. Cornice failures near profile GF3 in both April
2017 and March 2018 triggered small slab avalanches, but the majority of the avalanche debris resulted from entrainment as
the cornice blocks bounced downslope.

Platåberget's topography promotes slightly different avalanche dynamics. The gentler slope at the plateau edge allows snow
to accumulate directly under the cornices such that failed cornice masses land directly on the snow to be released as an
avalanche. Release areas on Platåberget collect snow during accretion events much more efficiently than those on
Gruvefjellet, where blowing snow mass losses due to suspension are promoted by the separation created between the
cornices and the release areas by the bedrock cliff. Accumulation in the upper release areas on Platåberget coinciding with
accretion events primes these locations for slab avalanche release with even small cornice failures. Relatively small cornice
failures triggering larger slab avalanches on Platåberget in April 2017 resulted in comparable magnitude avalanches (D2,
R2-R3) to those releasing from much larger cornice failures but less entrainable snow on Gruvefjellet in March 2018 (Figure
9b and Figure 13a).

## 5.3 Hazard management implications

Cornice fall avalanches are the most common avalanche type observed in the portion of central Svalbard surrounding our
study area where the broad plateau summits and steep valley walls of Longyeardalen's topography are recurrent across the
region (Eckerstorfer and Christiansen, 2011c). Cornice fall avalanches observed in this study thus represent processes
occurring elsewhere throughout central Svalbard – and to a lesser extent other locations throughout the world– and provide
an opportunity to reinforce existing forecasting frameworks with detailed cornice data. The conceptual model of avalanche
hazard in North America treats cornice failure both as an individual avalanche problem to be considered by forecasters and
as a potential trigger when assessing the likelihood of other avalanche types releasing in a given forecasting area and time
period (Statham et al., 2018). Cornice fall avalanche hazard assessments should thus consider both the likelihood of cornice
failure and the nature of the snow conditions in the release area to best judge cornice fall avalanche hazard. Our limited
dataset, especially in the absence of multiple air temperature induced failures, is insufficient to make broad generalizations
linking cornice failure type and resulting cornice fall avalanche activity. As a specific example, however, fairly widespread
wind slab avalanche activity throughout the region accompanied each of the accretion-induced avalanche events observed in
this work. The conditions leading to cornice accretion – strong winds and available snow for wind transport – also promote
the development of wind slab problems. Thus, when conditions are favorable for cornice accretion and accretion-induce
cornice failures, conditions are also favorable for the possible development of more widespread – and potentially more





sensitive – slab avalanche problems. In this scenario, the chance of a cornice failure triggering a secondary slab avalanche would rise, subsequently amplifying the cornice fall avalanche hazard by also increasing the expected size of the resulting cornice fall avalanche. Furthermore, in all cornice fall avalanches observed on Gruvefjellet the main cornice blocks travelled further downslope than the rest of the avalanche debris. This pattern is apparent on larger failures on Platåberget as well, but is in some cases less obvious, likely due to the smaller cornice blocks being functionally indistinguishable from the

avalanche debris. While the dataset presented here is insufficient to draw more quantitative conclusions regarding the runout distance of these cornice blocks, hazard management strategies should consider the destructive potential and extended runout of these blocks relative to the other entrained snow.

## 5.4 Uncertainties and limitations

The TLS data acquisition and processing techniques employed in this work allowed us to illustrate and quantify changes to

the observed cornice systems in detail not previously achieved, but our results and subsequent interpretations are nonetheless limited by several factors. Measurement uncertainties specifically related to measuring snow surfaces with TLS are well-discussed in previous research (Deems et al., 2015;Prokop, 2008), but we introduced additional uncertainty to our results and interpretations due to the scan timing. Our TLS data acquisition scheme involved time-intensive manual input, so we were unable to achieve the temporal resolution required to better constrain individual accretion and cornice failure events.

Decreasing time between scans would allow for more continuous and robust accretion rate calculations and could better constrain failure and avalanche snow surfaces, especially pre-event. Sufficiently decreasing the between-scan interval to a sub-daily resolution for such applications would likely require some degree of automation, and future work should consider employing a permanently-installed TLS acquiring data automatically similar to systems employed for mining applications or slope stability assessments.

Our experimental design focused on investigating the evolution and failure of the lower cornice surfaces from scan positions underneath the cornices where accessibility has precluded previous research. These scan positions did not, however, allow for systematic monitoring of the cornice roof. The orientation of the cornices' leading edges frequently shielded the cornice roof from the scanner, and our profiles often do not include the complete cornice roof. This also has implications for representative volume calculations, as uncertainty in the location of the cornice roof can result in inaccurate horizontal

difference calculations in these specific locations. By failing to capture the cornice roof in our data, we also limit comparisons with earlier work on Gruvefjellet relating downslope cornice deformation and cornice failure to the appearance of tension cracks between the cornice roof and the plateau anchoring point (Vogel et al., 2012). Future work should pair TLS data with some form of tension crack observation, and approaches combining TLS and UAV photogrammetry present intriguing possibilities for future work in this and other locations.

TLS was shown to be a particularly suitable remote sensing tool for cornice monitoring in Svalbard where we were able to obtain useful data during the early winter seasons when the polar night precludes direct visual observation and cornice



photography. Svalbard's unique environmental characteristics – such as the polar night – limit to a degree the applicability of our results to lower latitudes where more diurnal variations in radiation and temperature may influence cornice dynamics in ways not represented in Svalbard (e.g. Munroe, 2018). It is also unclear how representative the two winter seasons for which

we present data are for the cornice systems in Longyeardalen, as previous research has also noted considerable differences in cornice dynamics between seasons (Vogel et al., 2012). Continued cornice monitoring in this and other lower latitude settings would help clarify such uncertainties.

## 6 Conclusions

We monitored seasonal cornice dynamics and associated cornice fall avalanche activity with a TLS over two winter seasons

in high-Arctic Svalbard. The spatial and temporal resolution at which we acquired snow surface data with the TLS allowed us to quantify changes to the cornices with sub-decimeter accuracy. These data provide quantitative reinforcement to existing conceptual models of cornice dynamics and further strengthen the validity of these models. Notable quantitative contributions from this work include documentation of conservatively calculated horizontal accretion rates well in excess of 10 mm hr$^{-1}$ and a methodology for calculating cornice volumes from TLS data.

This study demonstrated the viability of TLS methods for monitoring cornice dynamics. TLS methods for obtaining snow surface data are suitable in Svalbard where the long polar night precludes data acquisition via other methods (e.g. photogrammetry), but techniques presented in this work are also suitable for cornices in other, lower latitude environments. Future work should investigate automated TLS data acquisition as an avenue to improve the temporal resolution of the measurements and better constrain cornice dynamics to specific meteorological conditions.

Our findings show complex interactions between topography, wind speed and direction, snow available for transport, existing snowpack, and cornice structure govern the growth, failure, and associated avalanche activity of the cornices in Longyeardalen. In particular, we show cornices rapidly accrete given winds strong enough to mobilize surface snow from a direction roughly perpendicular to the plateau edge, placing the cornices in the lee. Our findings also reinforce previous work indicating an increased likelihood of cornice failure and associated avalanche activity during these periods of cornice

accretion. This is encouraging for hazard managers seeking to forecast cornice fall avalanches, as anticipating the relatively infrequent conditions leading to cornice accretion can help predict periods of elevated cornice fall avalanche hazard. We observed the largest failures in our dataset in areas with minimal topographic support, demonstrating knowledge of the topography underlying the cornices can be beneficial when considering the specific location of cornice failure. Nevertheless, our limited dataset of cornice failures hinders conclusions drawn from this work, and continued work in a variety of

environments is needed to better understand the specific mechanisms and dynamics of cornice fall avalanches.



*Data Availability.* Data for the Svalbard Airport AWS are available through Norwegian Meteorological Institute's online data accessibility platform ([www.eklima.no](www.eklima.no)). Data from the Gruvefjellet AWS are freely available through the University Centre in Svalbard via [www.unis.no/resources/weather-stations/](www.unis.no/resources/weather-stations/). All TLS, snow depth, and other auxiliary data can be

obtained by contacting the corresponding author by email ([holt.hancock@unis.no](mailto:holt.hancock@unis.no)).

*Author contributions.* HH was responsible for the majority of the data acquisition, analyses, and interpretation of the results. ME helped develop the conceptual framework for the study and contextualized and interpreted the results within a broader snow and avalanche perspective. AP provided technical guidance with regards to TLS data acquisition and analysis techniques and assisted in the development of the study's technical framework in addition to assisting in data acquisition. JH

provided advice and supervision relating to study design, data analysis, and interpretation of the results. HH and ME were responsible for manuscript preparation with input from AP and JH.

*Competing interests.* The authors declare they have no conflicts of interest.





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



**Tables**

**Table 1.** Summary of avalanche cycles.

| Event date | Area | Trigger | Pre-event scan date | Post-event scan date | # observed cornice fall avalanches |
|---|---|---|---|---|---|
| 21 January 2017 | Platåberget | Accretion | 12 January 2017 | 22 January 2017 | >3 |
| 09 April 2017 | Gruvefjellet | Accretion | 21 March 2017 | 25 April 2017 | 1 |
| 21 April 2017 | Gruvefjellet | Accretion | 21 March 2017 | 25 April 2017 | 1 |
| 29 April 2017 | Platåberget | Accretion | 25 April 2017 | 01 May 2017 | >3 |
| 14 January 2018 | Gruvefjellet | Temperature (rain) | 15 December 2017 | 24 January 2018 | 1 |
| 18 March 2018 | Gruvefjellet | Accretion | 2 March 2018 | 23 March 2018 | 2 |

**Table 2.** Summary of well-constrained accretion events

| Area | Pre-event scan date and time (UTC) | Post-event scan date and time (UTC) | Between scan interval (hrs) | Profile with max horizontal accretion | Max horizontal accretion (m) | Accretion rate (mm hr$^{-1}$) |
|---|---|---|---|---|---|---|
| Platåberget | 12 January 2017 1930 | 21 January 2017 2100 | 217.5 | PB2 | 3.6 | 17 |
| Gruvefjellet | 17 February 2017 0900 | 24 February 2017 1100 | 170 | GF1 | 2.9 | 17 |
| Platåberget | 25 April 2017 1315 | 01 May 2017 0945 | 140.5 | PB1 | 2.0 | 14 |

**Table 3.** Seasonal summaries. All parameters are measured at the Gruvefjellet AWS except for precipitation, which is measured at the Svalbard Airport AWS.

| | 2016/2017 | 2017/2018 |
|---|---|---|
| Mean seasonal air temperature | -9.3 | -7.5 |
| Accumulated precipitation (mm) | 125.6 | 124.5 |
| % hours in season with accretion winds on Platåberget | 5.7 | 5.1 |
| % hours in season with accretion winds on Platåberget and daily precipitation > 0.2 mm | 3.0 | 1.7 |
| % hours in season with accretion winds on Gruvefjellet | 13.5 | 21.7 |
| % hours in season with accretion winds on Gruvefjellet and daily precipitation > 0.2 mm | 4.0 | 4.1 |
| Accretion winds on Platåberget: wind speed > 5 m s$^{-1}$, 225° < wind direction < 315°  Accretion winds on Gruvefjellet: wind speed > 5 m s$^{-1}$, 45° < wind direction < 135 | | |





**Figures**

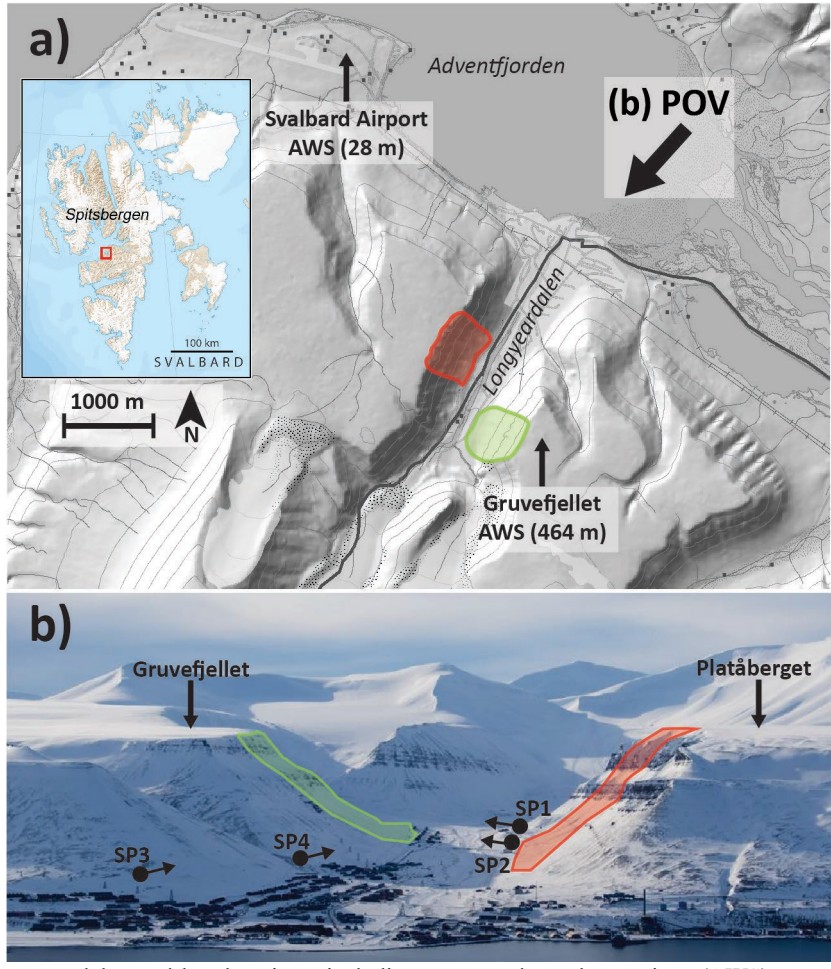

**Figure 1.** Overview of Longyeardalen and key locations, including automated weather stations (AWS), mentioned in the text. Contour lines in panel (a) are spaced at 100 m. The location and direction from which the photo in panel (b) was taken is indicated in (a). The location and extent of the Gruvefjellet and Platåberget study sites are indicated in (a) and (b) with green and red shading, respectively. Locations of scan positions SP1, SP2, SP3, and SP4 as well as the orientation of the scanner at each position are also indicated.




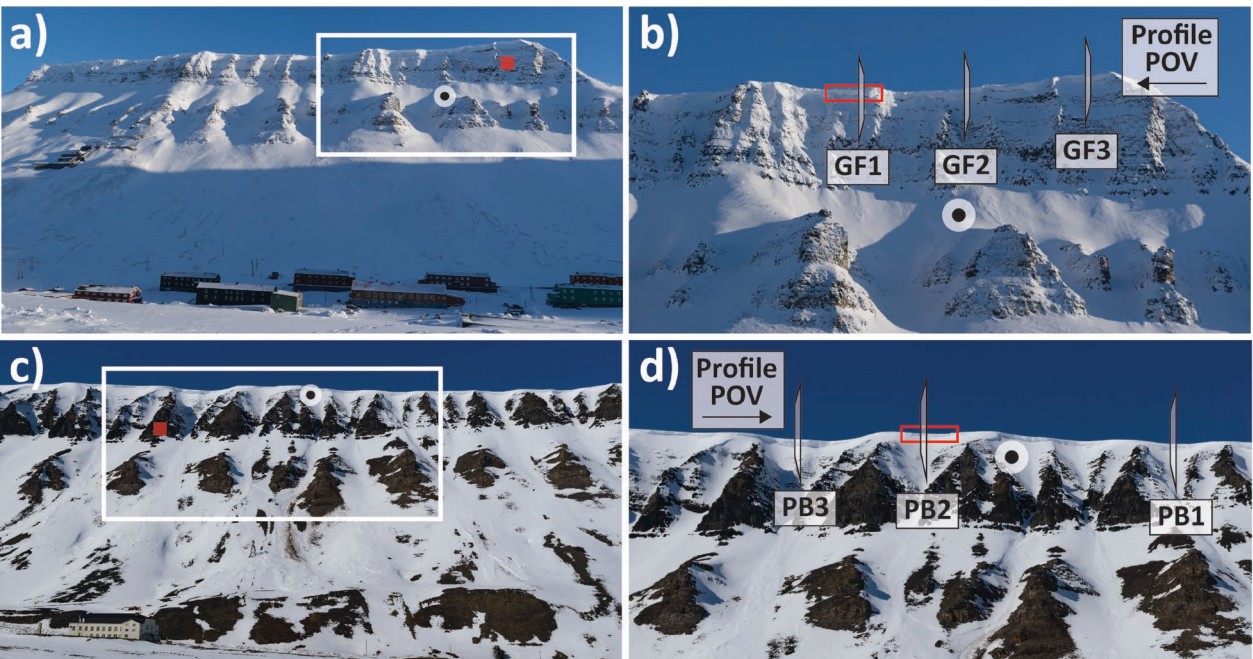

**Figure 2.** Overview of the cornice systems and locations of the primary spatial data employed in this work. Panel (a) shows Gruvefjellet from SP1 taken on 21 March 2017, with the white rectangle approximating the 600 m horizontal extent of panel (b) and the red square showing the area used to assess relative accuracy. Panel (b) indicates the location of the 2D cross-sectional profiles GF1, GF2, and GF3 in addition to outlining the area used to calculate representative cornice volume in red. Panel (c) shows Platåberget from SP4 taken on 24 May 2017, with the white rectangle approximating the 600 m horizontal extent of panel (d) and the red square showing the area used to assess relative accuracy. Panel (d) indicates the location of the 2D cross-sectional profiles PB1, PB2, and PB3 in addition to outlining the area used to calculate representative cornice volume in red. Snow depth sensor locations are indicated in all panels with the black dot.



**Figure 3.** Visualization of the point cloud processing methods in CloudCompare. Panel (a) shows a photo of the cornice represented by profile GF1 on 21 March 2017. Panel (b) shows the same surface as represented by the 0.10 m point cloud. The manually defined axis of
GF1 is indicated by the white line. Panels (c) and (d) show the 21 March scanned surface and extracted profile from two vantage points. Panel (e) displays both the 21 March (colored points) and bare-earth (white points) surfaces oriented parallel to the projection direction, with the 21 March profile (green) and bare-earth profile (white) also indicated. Panels (f) and (h) display similar data but with the surfaces oriented roughly perpendicular to the projection direction (shown with red arrows), and panel (h) shows a cross-section of the surface shown in panels (e) and (f). The 1 m grid showing the horizontal differences between the 21 March and bare-earth scan is displayed in
panel (g). All scale bars are in meters

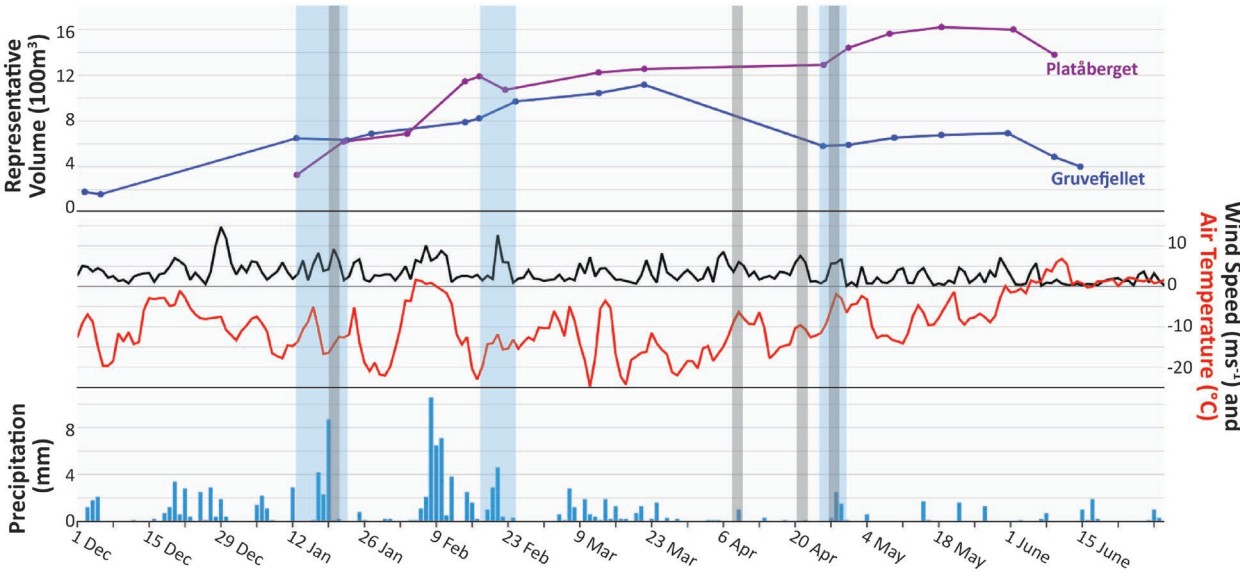

**Figure 4.** Summary of the representative cornice volume progression and meteorological conditions for the 2016/2017 winter season. Wind speed and air temperature are daily averaged values from the Gruvefjellet AWS, and precipitation data are daily values from the Svalbard Airport AWS measured at 0600 UTC. Shaded blue vertical bars indicate well-constrained cornice accretion periods for which we were able to calculate horizontal cornice accretion rates (Table 2). Shaded grey vertical bars indicate 48 hour periods with observed noteworthy cornice fall avalanche activity (Table 1).


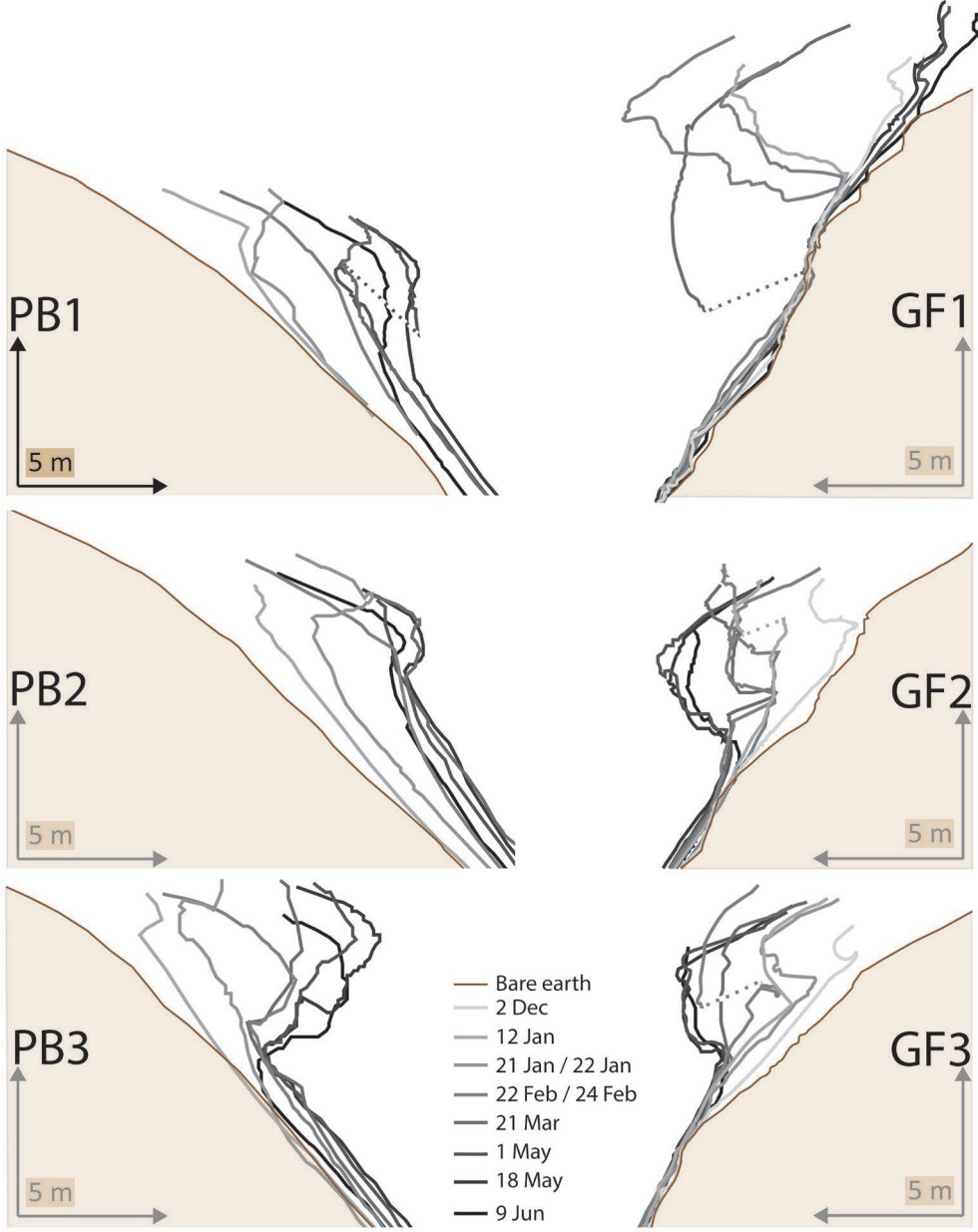

**Figure 5.** 2D cornice profiles showing cornice progression for selected scan dates throughout the 2016/2017 winter season. Each profile is labeled as it referred to in the text and corresponds to the location and POV depicted in Figure 2





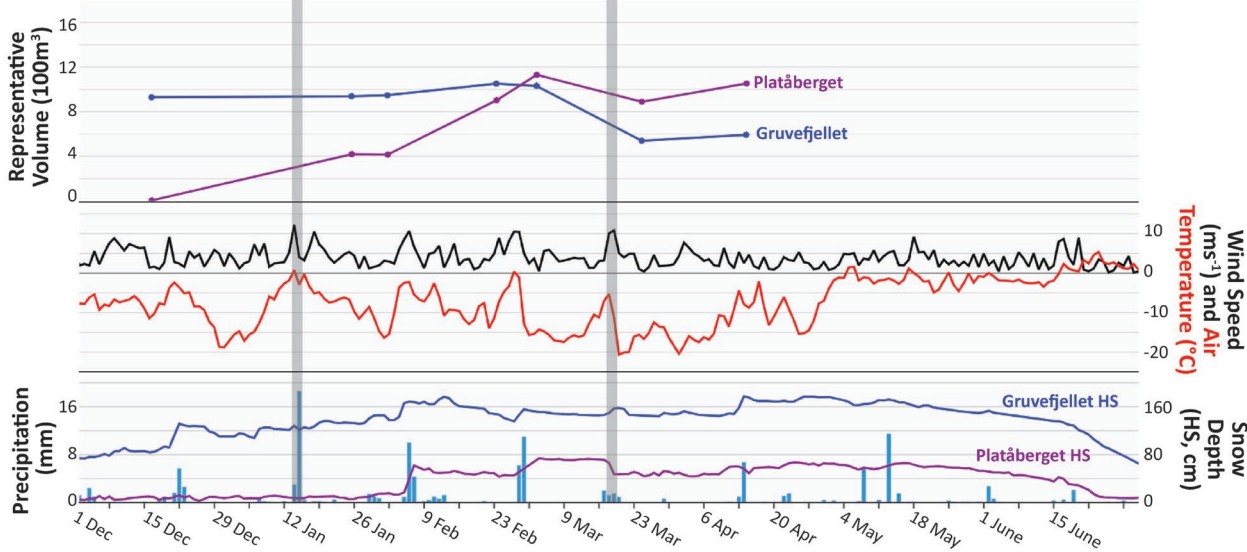


**Figure 6.** Summary of the representative cornice volumes and meteorological conditions for the 2017/2018 winter season. Wind speed and air temperature are daily averaged values from the Gruvefjellet AWS, precipitation data are daily values from the Svalbard Airport AWS measured at 0600 UTC, and snow depths are daily averages from the snow sensors on Gruvefjellet and Platåfjellet. Shaded grey vertical bars indicate 48 hour periods with observed noteworthy cornice fall avalanche activity (Table 2).




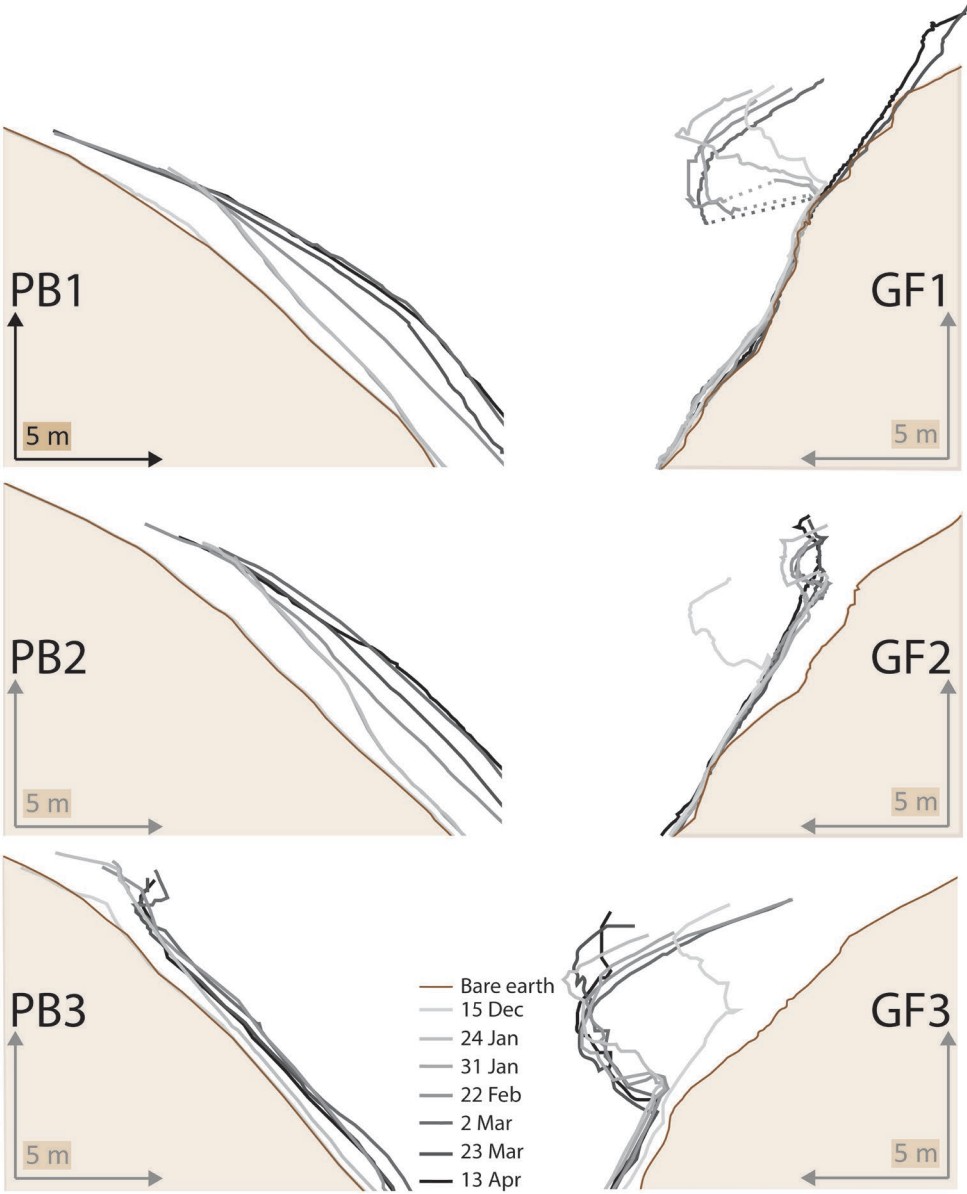


**Figure 7.** 2D cornice profiles showing cornice progression for 2017/2018 winter season scan dates. Each profile is labeled as it referred to in the text and corresponds to the location and POV depicted in Figure 2.




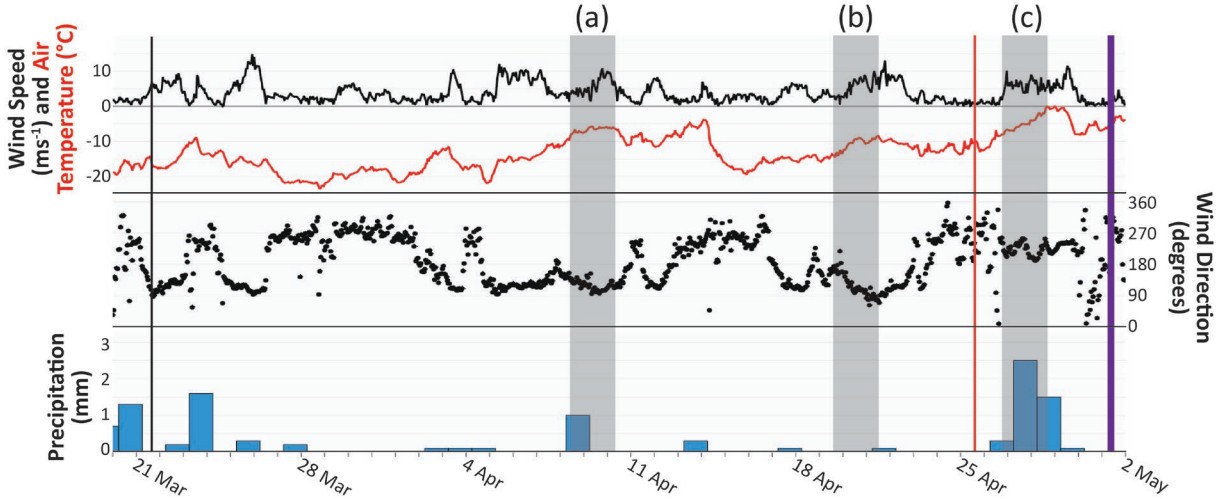

**Figure 8.** Meteorological summary of the April 2017 case study. Wind speed, wind direction, and air temperature are hourly values from the Gruvefjellet AWS, and precipitation data are daily values from the Svalbard Airport AWS measured at 0600 UTC. Colored vertical lines in the time series indicate the scan timing corresponding to the profiles in Figure 9. Vertical grey bars marked (a), (b), and (c) correspond to 48 hour time periods with noteworthy avalanche activity discussed in the text.




**Figure 9.** M3C2 distances displaying changes to the snow cover on Gruvefjellet between the 21 March and 25 April 2018 scans (a) and on
Platåberget between the 25 April and 1 May 2018 scans (b). Red rectangles in both panels indicate the locations of the cornice profiles.
Specific snow surface features are annotated as they are referred to in the text.

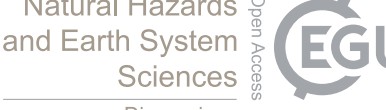

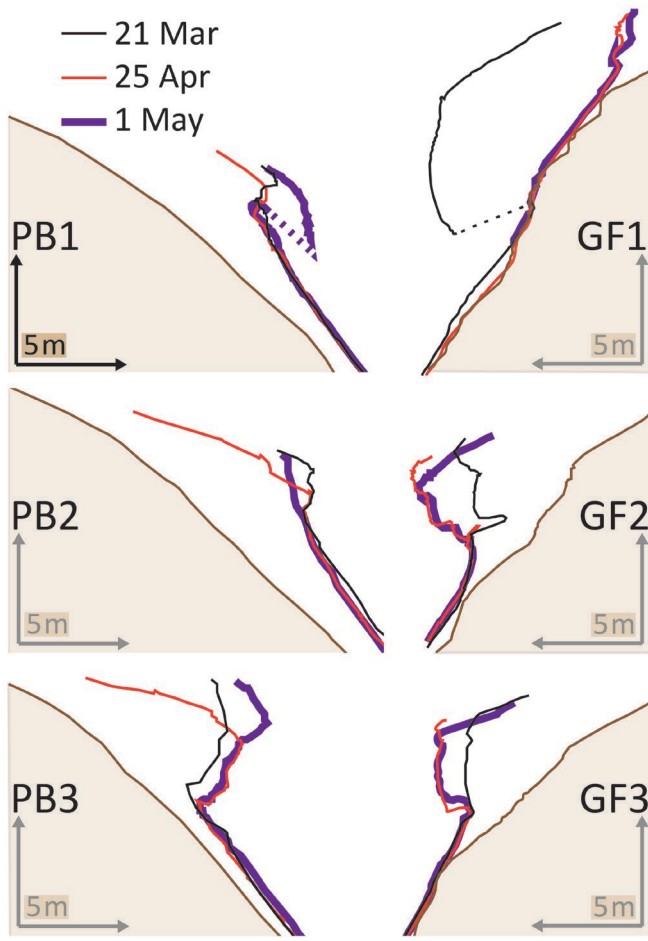

**Figure 10.** Cornices profiles illustrating cornice dynamics during the April 2017 case study, with each profile labeled as it is referred to in the text. Dashed lines indicate interpolated data where overhanging cornice structure shadowed the snow surface from the TLS.



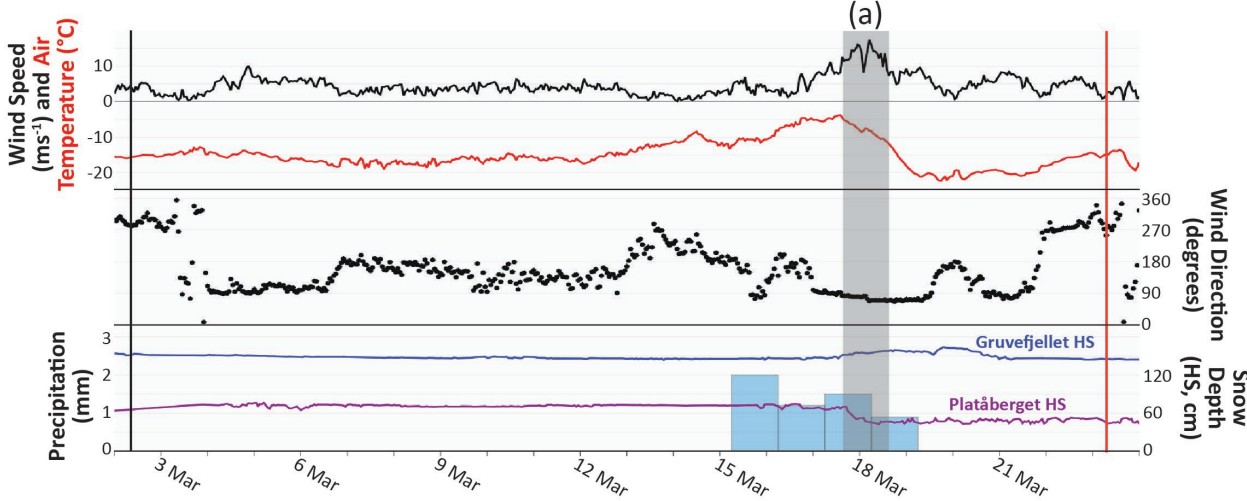


**Figure 11.** Meteorological summary of the March 2018 case study. Wind speed, wind direction, and temperature are hourly values from the Gruvefjellet AWS, and precipitation data are daily values from the Svalbard Airport AWS measured at 0600 UTC. Colored vertical lines in the time series indicate the scan timing corresponding to the profiles in Figure 12, and the grey vertical bar annotated with (a) corresponds to the 48 hour time period with noteworthy avalanche activity mentioned in the text.



**Figure 12.** M3C2 distances displaying changes to the snow cover on Gruvefjellet (a) and Platåberget (b) between the 2 March and 23 March 2018 scans. Red rectangles in both panels indicate the locations of the cornice profiles. Specific snow surface features are annotated as they are referred to in the text, and snow depth sensors are marked and labeled.





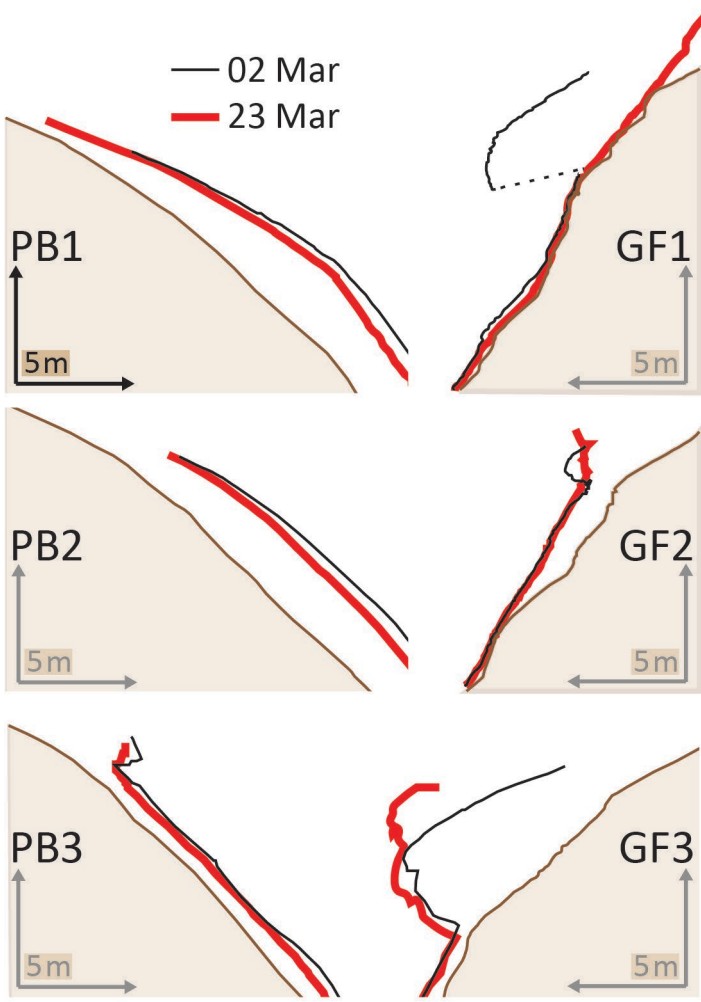

**Figure 13.** Cornices profiles illustrating cornice dynamics during the March 2018 case study, with each profile labeled as it is referred to in the text. Dashed lines indicate interpolated data where overhanging cornice structure shadowed the snow surface from the TLS.



## Appendix 1

| Date and Time (UTC) | Area | Scan Position | Reported Registration Error | Mean Relative Error (m) | Standard Deviation (m) | Representative Volume (m³) | Volume Uncertainty |
|---|---|---|---|---|---|---|---|
| 16 September 2016 12:00 | Gruvefjellet | S1 | NA | NA | NA | NA | NA |
| 16 September 2016 13:00 | Platåberget | S3 | NA | NA | NA | NA | NA |
| 02 December 2016 10:15 | Gruvefjellet | S1 | 0.003 | 0.036 | 0.001 | 179 | 13.173 |
| 05 December 2016 12:50 | Gruvefjellet | S1 | 0.023 | 0.019 | 0.016 | 159 | 6.827 |
| 12 January 2017 17:30 | Gruvefjellet | S4 | 0.002 | 0.047 | 0.028 | 649 | 17.343 |
| 12 January 2017 19:30 | Platåberget | S3 | 0.002 | 0.067 | 0.039 | 328 | 24.612 |
| 21 January 2017 21:00 | Platåberget | S3 | 0.004 | 0.032 | 0.044 | 619 | 11.956 |
| 22 January 2017 14:00 | Gruvefjellet | S2 | 0.030 | 0.030 | 0.025 | 633 | 11.144 |
| 27 January 2017 08:50 | Gruvefjellet | S1 | 0.020 | 0.024 | 0.016 | 689 | 8.930 |
| 03 February 2017 08:00 | Platåberget | S4 | 0.004 | 0.000 | 0.044 | 688 | 0.148 |
| 14 February 2017 14:00 | Platåberget | S4 | 0.018 | 0.066 | 0.039 | 1145 | 24.170 |
| 14 February 2017 15:00 | Gruvefjellet | S1 | 0.003 | 0.022 | 0.020 | 789 | 7.970 |
| 17 February 2017 09:00 | Gruvefjellet | S1 | 0.002 | 0.032 | 0.030 | 823 | 11.771 |
| 17 February 2017 10:00 | Platåberget | S4 | 0.004 | 0.055 | 0.039 | 1190 | 20.111 |
| 22 February 2017 10:45 | Platåberget | S3 | 0.035 | 0.042 | 0.037 | 1072 | 15.350 |
| 24 February 2017 11:00 | Gruvefjellet | S1 | 0.003 | 0.027 | 0.031 | 970 | 9.926 |
| 12 March 2017 16:00 | Platåberget | S4 | 0.005 | 0.003 | 0.046 | 1224 | 1.033 |
| 12 March 2017 17:30 | Gruvefjellet | S2 | 0.013 | 0.034 | 0.027 | 1043 | 12.435 |
| 21 March 2017 13:10 | Gruvefjellet | S1 | 0.014 | 0.031 | 0.027 | 1117 | 11.402 |
| 21 March 2017 14:10 | Platåberget | S4 | 0.017 | 0.064 | 0.042 | 1255 | 23.616 |
| 25 April 2017 10:00 | Gruvefjellet | S1 | 0.035 | 0.068 | 0.027 | 581 | 25.018 |
| 25 April 2017 13:15 | Platåberget | S4 | 0.019 | 0.007 | 0.042 | 1291 | 2.731 |
| 01 May 2017 09:45 | Platåberget | S4 | 0.021 | 0.045 | 0.071 | 1440 | 16.753 |
| 01 May 2017 10:25 | Gruvefjellet | S1 | 0.028 | 0.034 | 0.020 | 591 | 12.472 |
| 08 May 2017 10:15 | Platåberget | S4 | 0.025 | 0.015 | 0.045 | 1563 | 5.535 |
| 09 May 2017 08:25 | Gruvefjellet | S1 | 0.008 | 0.041 | 0.023 | 654 | 15.166 |
| 18 May 2017 12:05 | Gruvefjellet | S1 | 0.022 | 0.076 | 0.016 | 677 | 28.044 |
| 18 May 2017 13:00 | Platåberget | S4 | 0.007 | 0.011 | 0.039 | 1620 | 3.948 |
| 31 May 2017 10:40 | Gruvefjellet | S1 | 0.010 | 0.061 | 0.020 | 693 | 22.546 |
| 01 June 2017 13:15 | Platåberget | S4 | 0.020 | 0.033 | 0.062 | 1599 | 11.993 |
| 09 June 2017 12:20 | Gruvefjellet | S1 | 0.008 | 0.018 | 0.015 | 485 | 6.642 |
| 09 June 2017 13:30 | Platåberget | S4 | 0.013 | 0.014 | 0.039 | 1379 | 5.203 |
| 14 June 2017 14:35 | Gruvefjellet | S1 | 0.003 | 0.005 | 0.010 | 401 | 1.845 |



*Appendix 1 continued…*

| Date and Time (UTC) | Area | Scan Position | Reported Registration Error | Mean Relative Error (+- m) | Standard Deviation | Representative Volume (m³) | Volume Uncertainty |
|---|---|---|---|---|---|---|---|
| 15 December 2017 10:20 | Gruvefjellet | S1 | 0.013 | 0.018 | 0.024 | 929 | 6.458 |
| 15 December 2017 11:00 | Platåberget | S4 | 0.060 | 0.016 | 0.039 | 5 | 5.978 |
| 24 January 2018 11:20 | Gruvefjellet | S1 | 0.000 | 0.049 | 0.031 | 938 | 18.081 |
| 24 January 2018 12:25 | Platåberget | S4 | 0.032 | 0.069 | 0.062 | 419 | 25.277 |
| 31 January 2018 15:45 | Gruvefjellet | S1 | 0.025 | 0.009 | 0.024 | 947 | 3.321 |
| 31 January 2018 17:00 | Platåberget | S4 | 0.044 | 0.070 | 0.052 | 415 | 25.978 |
| 22 February 2018 10:30 | Gruvefjellet | S1 | 0.017 | 0.020 | 0.025 | 1051 | 7.528 |
| 22 February 2018 11:30 | Platåberget | S4 | 0.067 | 0.032 | 0.056 | 902 | 11.734 |
| 02 March 2018 11:55 | Gruvefjellet | S1 | 0.004 | 0.003 | 0.037 | 1031 | 1.144 |
| 02 March 2018 12:30 | Platåberget | S4 | 0.001 | 0.024 | 0.072 | 1130 | 8.819 |
| 23 March 2018 13:15 | Gruvefjellet | S1 | 0.001 | 0.041 | 0.023 | 539 | 15.092 |
| 23 March 2018 14:00 | Platåberget | S4 | 0.013 | 0.032 | 0.049 | 889 | 11.845 |
| 13 April 2018 11:00 | Gruvefjellet | S1 | 0.012 | 0.024 | 0.021 | 593 | 8.745 |
| 13 April 2018 13:20 | Platåberget | S4 | 0.010 | 0.000 | 0.044 | 1053 | 0.037 |