# Peer review of "QUANTIFYING SEASONAL CORNICE DYNAMICS USING A TERRESTRIAL LASER SCANNER IN SVALBARD, NORWAY"

_Natural Hazards and Earth System Sciences, 2019_

## Referee Comment (RC1) · Christine Fey (Referee) · 20 Nov 2019

The paper deals with the TLS based analyses of the cornice processes. To my Knowledge, yet there are no other studies on cornice analyses by TLS. Since I am more expert on laser scanning and landslide processes than on snow research I can evaluate the methodological part of the paper in detail. However, the data interpretation and cornice process analyses seem to reasonable, comprehensible and easy to follow. The paper is very well written, structured and clear figures supplement the text. I suggest publishing the paper after minor revisions.

I added some commentaries to the pdf. Here some further suggestion concerning laser scanning:

[Figure]

It would be helpful to add the tiepoints used for registration of the scans to the figures. To assess the accuracy only one area is used. The registration error can vary significant between different areas. Because of this, the mean relative error is not representative for the entire scan. Either the authors enhance the accuracy assessment to more stable areas, which might be difficult in the case that there are now other snow free areas. The other option is, to explain in the text why only one area was used. However, the research question focuses on changes in the magnitude of meters and in this case the accuracy assessment is not so important for the process understanding of cornices. More critical I evaluate the volume estimation of cornices. Here, i) the TLS data uncertainties play a more important role in quantification and ii) it's very challenging to calculate the distance and volume of cornices at scoured areas iii) data gaps are causing significant uncertainties in volume quantification. The problem of data gaps is mentioned and the other points could be enhanced in the discussion. Since the process interpretation in this paper is mainly based on the shape of cornices taken from 2D profiles and distance changes of the snowpack it might be better to study the volume calculation of cornices in detail and publish it in an another research work.

Please also note the supplement to this comment:
https://www.nat-hazards-earth-syst-sci-discuss.net/nhess-2019-329/nhess-2019-329-RC1-supplement.pdf

---

## Referee Comment (RC2) · Jeffrey Munroe (Referee) · 6 Dec 2019

General Comments: I appreciate the opportunity to review this well-written manuscript presenting an intriguing use of terrestrial laser scanning (TLS) technology. The authors worked in Longyearbyen, Svalbard where an important body of prior work has focused on cornice fall avalanches. The innovative addition here is the TLS, which allowed more accurate volume estimations to be made for cornice on slopes facing two different aspects, over parts of two consecutive winters. Sustained winter darkness at this latitude complicates conventional methods like visual observation or time-lapse photography, but the TLS overcomes limitations of daylight. The long-range scanner employed here also allows slopes to be repeatedly measured from a safe distance, while the use of snow-free, bedrock surfaces as reference points improves the accuracy of the snow

volume calculations. Cornice fall avalanches are important hazards in this and other mountainous regions, but considerable uncertainty remains about the processes and triggers responsible for these events. As such, this work has important ramifications for planning and hazard management. I have no doubt that the international snow science community will be interested in this application of TLS. I also expect that this work will evolve rapidly in future years, likely incorporating greater automation to allow cornice growth/fail to be tracked over much finer temporal scales. I find the manuscript to be well organized and clearly written. By working carefully over two winters, on two different slopes, the authors were able to identify consistencies with implications for understanding the processes involved in cornice formation. The title is accurate. The methods are clearly explained. The literature review and introduction are concise, but adequately present the motivation for the study. The overall presentation is clearly and logically structured, and the paper is an appropriate length. I recommend acceptance after a few minor changes (noted below).

Specific Comments: All of the figures are relevant and helpful to the reader, but I think a few of them should be changed. Specifically, Figure 5 and 7 are difficult to read because the colors used to represent the snow surface at the different times are too similar. Both of these figures are really important – they nicely present the data and allow clear visual distinctions to be made between scans, between slopes, and between the two winters. Improving their readability with more contrasting colors is a necessary step that will greatly help comprehension of the reader. I found Figures 9 and 12 difficult for a similar reason. While one might think that the bright red and dark blue colors representing the extremes of the change spectrum would be visible against the grayscale hillshade, the differences are actually really subtle. The figures are both important because they illustrate just how sensitive the TLS method is to even small changes in the snow surface. Unfortunately, the areas that changed are just really hard to see – even with the arrows drawing attention to specific regions in the images. I'm not sure what to recommend here. It is possible that a different color scheme would work better. Another possibility would be to keep a large figure representing the overview

[Figure]

of the slope, and having a series of enlargements of small areas (keyed back to boxes in the overview figure) that show the detected changes in a more obvious, zoomed-in way. Just as in my comment above, these figures are critical to presenting your data and supporting the following discussion – it would be great if they could be made even more compelling.

Technical Corrections: After reading through the manuscript, I also offer the following minor editorial suggestions: Line 30 – ". . . projections of snow that form due to. . ." Line 40 – ". . . Cornice hazards." Line 76 – Would "designing" be a better word than "planning" here? Line 90 – I usually capitalize "U-shaped valley" Line 90 – ". . . oriented axis running. . ." Line 95 – Is there any information about the thickness of the continuous permafrost? Line 101 – ". . . consists of a 50-70-m, near-vertical bedrock cliff situated under the plateau margin and above. . ." Line 115 – "The climate of Svalbard prohibits. . ." Line 139 – by "reliable" snow depth data do you mean the start of seasonal snow accumulation? Or is this the date at which the snowpack exceeded a certain minimum thickness necessary for accurate measurement? Line 154 – ". . . we used to georeference individual. . ." Line 225 – I'm not sure what "(Size D2, R3")" means. Line 370-372 – I would include reference to Figure 7 and Figure 5 here. Line 380 – ". . . to suggest that specific interactions. . ." Line 387 – ". . . cornices we investigated, and also failed completely both seasons." Line 439 – ". . . also favorable for the development of more. . ." Line 629 – ". . . was taken is indicated by POV in. . ."

---

## Author Comment (AC1) · 17 Jan 2020

**Response to RC1**

We appreciate Dr. Christine Fey for reviewing our work and for her constructive comments to improve this manuscript. We first respond to issues and suggestions provided directly in Dr. Fey's interactive comment and subsequently respond to comments and corrections included in Dr. Fey's supplement to the interactive comment. Reviewer comments are displayed below in bold, author responses are in standard text.

**Responses to the interactive comment:**

**It would be helpful to add the tiepoints used for registration of the scans to the figures.**

We agree visual tiepoint – and tieobject – display in the figures would enhance this work. Unfortunately, we were unable to use the same tiepoints and tieobjects to register each of the scans. This is primarily due to changing snow cover rendering certain tiepoints and tieobjects unusable as snow free surfaces at different times throughout the two winter seasons. Thus, we drew on a suite of over 50 tieobjects to adequately register all scans included in this work, and we believe including all tiepoints and tieobjects we employed would overly clutter the figures. Additionally, some of the tiepoints were located on buildings not included in the figures' extent.

**To assess the accuracy only one area is used. The registration error can vary significantly between different areas. Because of this, the mean relative error is not representative for the entire scan. Either the authors enhance the accuracy assessment to more stable areas, which might be difficult in the case that there are no other snow free areas. The other option is to explain in the text why only one area was used. However, the research question focuses on changes in the magnitude of meters and in this case the accuracy assessment is not so important for the process understanding of cornices.**

We struggled with this issue ourselves. The concern again, to which Dr. Fey alludes in her review, is the dearth of adequately large snow free areas in the vicinity of the cornices with which to use for accuracy assessments. One option would have been to report accuracies from building walls near the foot of the slope. Accuracy near these buildings is generally very good – and almost always better than near the cornices, as the building roofs and walls served as good surfaces to use in the registration process. Since we agree with Dr. Fey that the accuracy assessment is not so important for the process understanding of cornices, we elected to display accuracy assessment from the only stable, snow free area near the cornices because:

- Accuracy assessments from this location are most representative of scan accuracies near the cornices, and
- Reported accuracies from this location are typically worse than in other locations in the scanning domain (i.e. the relative accuracies assessed from building walls would not be representative of the accuracy near the cornices).

As suggested by Dr. Fey, we have added language in lines 195 – 196 clarifying our decision to use just a single area for accuracy assessments.

**More critical I evaluate the volume estimation of cornices. Here, i) the TLS data uncertainties play a more important role in quantification and ii) it's very challenging to calculate the distance and volume of cornices at scoured areas iii) data gaps are causing significant uncertainties in volume quantification. The problem of data gaps is mentioned and the other points could be enhanced in the discussion. Since the process interpretation in this paper is mainly based on the shape of cornices taken from 2D profiles and distance changes of the snowpack it might be better to study the volume calculation of cornices in detail and publish in another research work.**

All three of the issues raised with our volume estimation of cornices are valid concerns. We have specifically addressed these concerns in lines 464 – 471 of the discussion rather than eliminating these analyses altogether from the paper. We agree another research work specifically studying cornice volume changes is needed, but based on the data we currently have gathered and Reviewer 2's positive responses we have chosen to leave these analyses in this work, acknowledging the uncertainty in our calculations.

**Responses to the supplement to the comment:**

*Minor grammatical comments and suggestions have been corrected as suggested by the reviewer.*

Revisions from the supplement affecting the content of the manuscript:

**Line 156 – The values given from the RiScan MSA are not the 3D accuracy and does only reflect the distance between the point correspondences used at the ICP. I would neglect this value because it says nothing about the registration quality.**

True, and we did not explain this adequately. Based on these recommendations, however, we will remove both this sentence and the MSA values from Appendix 1.

**Line 158 – You mean the point cloud was thinned by blockthinning? The term "grid" is misleading with raster data.**

We've updated the sentence with more specific and less misleading language. Thank you.

**Line 207 – Headings in chapter 4 can be more meaningful which would help the reader to understand the structure of the paper faster.**

We've updated the headings in chapter 4 to "Seasonal summaries of meteorological conditions and cornice dynamics", which will hopefully help with readability and clarity.

**Line 635 – A legend would be helpful for faster understanding of the figure.**

We've added a legend for the symbology where appropriate and removed redundant language from the figure caption.

**Lines 663 and 667 – Revise the color scheme of the profile to distinguish between the dates.**
We've changed the color scheme of the profiles to hopefully enhance readability.

**Lines 683 and 687 – Revise the color scheme of the distance change to allow a better interpretation of the magnitude of change.**

We've adjusted the figure to make interpretation of the changes to the snow surface on the slope below the cornices easier. See the detailed response to RC2 regarding Figures 9 and 12.

---

## Author Comment (AC2) · 17 Jan 2020

**Response to RC2**

We are grateful for Dr. Jeffery Munroe's review and constructive comments to improve this manuscript. We respond to his comments below. Reviewer comments are displayed below in bold, author responses are in standard text.

**Specific comments:**

**All of the figures are relevant and helpful to the reader, but I think a few of them should be changed. Specifically, Figures 5 and 7 are difficult to read because the colors used to represent the snow surface at the different times are too similar. Both of these figures are really important – they nicely present the data and allow clear visual distinctions to be made between scans, between slopes, and between the two winters. Improving their readability with more contrasting colors is a necessary step that will greatly help comprehension of the reader.**

Thanks. We have struggled with the color schemes for this figure for a while. To address these concerns, we have attempted to render the diagrams with more contrasting colors than the greyscale we had originally selected. Although we acknowledge readability is still not perfect, we thought it was important to include all selected profiles in the seasonal profile progressions rather than eliminate some profiles in the interest of legibility.

**I found Figures 9 and 12 difficult for a similar reason. While one might think that the bright red and dark blue colors representing the extremes of the change spectrum would be visible against the grayscale hilshade, the differences are actually really subtle. The figures are both important because they illustrate just how sensitive the TLS method is to even small changes in the snow surface. Unfortunately, the areas that changed are just really hard to see – even with the arrows drawing attention to specific regions in the images. I'm not sure what to recommend here. It is possible that a different color scheme would work better. Another possibility would be to keep a large figure representing the overview of the slope, and having a series of enlargements of small areas (keyed back to boxes in the overview figure) that showed the detected changes in a more obvious, zoomed-in way. Just as in my comment above, these figures are critical to presenting your data and supporting the following discussion – it would be great if they could be made even more compelling.**

We agree the changes were too difficult to discern with the color scale as originally presented in the figure. The issue was attempting to display more subtle changes to the snow surface on the slopes below the cornices with a quantitative M3C2 distance scale suitable to represent the much greater changes to the cornices themselves (scale from -5 m to 5 m). To address this issue, we have changed the scale from -2 m to 2 m to better represent the changes to slope below the cornices. This hopefully helps differentiation between specific snow surface features, while also drawing attention to the dramatic changes on the cornices. We've added "greater than" or "less than" symbology to the scale labels to show that M3C2 distance changes on the cornices exceed the variability captured by this scale (also evident in the text and Figures 5 and 7).

**Technical corrections:**

**Line 30 – "…projections of snow that form due to…"**
Changed.

**Line 40 – "…Cornice hazards."**
Fixed.

**Line 76 – Would "designing" be a better word than "planning" here?**
Yes – we altered the sentence to hopefully improve clarity.

**Line 90 – I usually capitalize "U-shaped valley."**
We have changed the punctuation to reflect this.

**Line 90 – "…oriented axis running…"**
Changed, thank you.

**Line 95 – Is there any information about the thickness of the continuous permafrost?**
Yes, thank you for pointing out this omission. We've added the information from Humlum et al. (2003).

**Line 101 – "consists of a 50-70 m, near-vertical bedrock cliff situated under the plateau margin and above…"**
Thanks.

**Line 115 – "The climate of Svalbard prohibits…"**
Changed.

**Line 139 – by "reliable" snow depth data do you mean the start of the seasonal snow accumulation? Or is this the date at which the snowpack exceeded a certain minimum thickness necessary for accurate measurement?**
We meant the date at which the snowpack became deep enough to overcome to local ground surface roughness and become "smooth" enough to generate reliable measurements. Since this point is largely irrelevant for the larger purposes of this paper, we have removed the word "reliable" from the sentence to avoid confusion.

**Line 154 – "…we used to georeferenced individual…"**
Thanks.

**Line 225 – I'm not sure what "(Size D2, R3)" means.**
This refers to the destructive / relative size classifications used to characterize avalanche sizes. We've left this description in for now, but it can be easily removed if the editor finds it confusing or detrimental to the manuscript's clarity. We have also added a reference to the recording standards for avalanche sizes.

**Line 370-372 – I would include reference to Figure 7 and Figure 5 here.**
Good point, thanks. Figure references added.

**Line 380 – "…to suggest that specific interactions…"**
Changed.

**Line 387 – "…cornices we investigated, and also failed completely both seasons."**
Changed, thanks.

**Line 439 – "…also favorable for the development of more…"**
Thanks.

**Line 629 – "…was taken is indicated by POV in …"**
Fixed, thank you!